# Estimating Cotton Yield in the Brazilian Cerrado Using Linear Regression Models from MODIS Vegetation Index Time Series

**Daniel A. B. de Siqueira** [1] , **Carlos M. P. Vaz** [2,*] , **Flávio S. da Silva** [1] , **Ednaldo J. Ferreira** [2] , **Eduardo A. Speranza** [3] , **Júlio C. Franchini** [4] , **Rafael Galbieri** [5] , **Jean L. Belot** [5] , **Márcio de Souza** [5] , **Fabiano J. Perina** [6] and **Sérgio das Chagas** [7]

1 Central Paulista University Center, São Carlos 13563-470, SP, Brazil; danielbotta46@gmail.com (D.A.B.d.S.); fl_santos02@hotmail.com (F.S.d.S.)
2 Brazilian Agricultural Research Corporation, Embrapa Instrumentation, São Carlos 13561-354, SP, Brazil; ednaldo.ferreira@embrapa.br
3 Brazilian Agricultural Research Corporation, Embrapa Digital Agriculture, Campinas 13083-886, SP, Brazil; eduardo.speranza@embrapa.br
4 Brazilian Agricultural Research Corporation, Embrapa Soybean, Londrina 86001-970, PR, Brazil; julio.ranchini@embrapa.br
5 Mato Grosso Cotton Institute, Cuiabá 78049-015, MT, Brazil; rafaelgalbieri@imamt.org.br (R.G.); jeanbelot@imamt.org.br (J.L.B.); marciosouza@imamt.org.br (M.d.S.)
6 Brazilian Agricultural Research Corporation, Embrapa Cotton, Campina Grande 58428-095, PB, Brazil; fabiano.perina@embrapa.br
7 Amaggi Group, Sapezal 78365-000, MT, Brazil; sergio.chagas@amaggi.com.br
* Correspondence: carlos.vaz@embrapa.br; Tel.: +55-16-2107-2852

**Abstract:** Satellite remote sensing data expedite crop yield estimation, offering valuable insights for farmers' decision making. Recent forecasting methods, particularly those utilizing machine learning algorithms like Random Forest and Artificial Neural Networks, show promise. However, challenges such as validation performances, large volume of data, and the inherent complexity and inexplicability of these models hinder their widespread adoption. This paper presents a simpler approach, employing linear regression models fitted from vegetation indices (VIs) extracted from MODIS sensor data on the Terra and Aqua satellites. The aim is to forecast cotton yields in key areas of the Brazilian Cerrado. Using data from 281 commercial production plots, models were trained (167 plots) and tested (114 plots), relating seed cotton yield to nine commonly used VIs averaged over 15-day intervals. Among the evaluated VIs, Enhanced Vegetation Index (EVI) and Triangular Vegetation Index (TVI) exhibited the lowest root mean square errors (RMSE) and the highest determination coefficients ($R^2$). Optimal periods for in-season yield prediction fell between 90 and 105 to 135 and 150 days after sowing (DAS), corresponding to key phenological phases such as boll development, open boll, and fiber maturation, with the lowest RMSE of about 750 kg ha$^{-1}$ and $R^2$ of 0.70. The best forecasts for early crop stages were provided by models at the peaks (maximum value of the VI time series) for EVI and TVI, which occurred around 80–90 DAS. The proposed approach makes the yield predictability more inferable along the crop time series just by providing sowing dates, contour maps, and their respective VIs.

**Keywords:** remote sensing; cotton production forecast; Brazil

## 1. Introduction

Cotton holds significant economic importance in Brazil, ranking as the world's fourth largest producer and the second largest exporter. In the 2022–2023 season, Brazil cultivated cotton across 1.66 million hectares, resulting in a total lint production of 3.03 million tons [1].

The majority of the cotton cultivation occurs within the Brazilian Cerrado, characterized by flat and arable lands with favorable rainfall ranging from 750 to 2000 mm/year across the biome [2]. This region employs a highly mechanized, rainfed production system, with approximately 92% occurring without irrigation [3]. Typically, cotton is cultivated

from January to July, following the soybean harvest [4]. The primary cotton producing states are Mato Grosso and Bahia, jointly contributing to around 90% of Brazil's total output [5].

Estimating in-season cotton yield over large areas at a regional or national level provides critical information for farmers, policymakers, governments, crop insurers, and commodity traders. Yield estimations involve a combination of field surveys, remote sensing, statistical regression, and crop simulation models [6,7]. Mid-season crop forecasting is particularly vital for farmers, as it informs management decisions regarding harvesting, storage, transport logistics, and planning for subsequent crops [8,9]. Early forecasting cotton yield, before 90–100 days after sowing, provides valuable management information for implementing corrective interventions, such as adjusting input applications (e.g., top-dressing fertilizer and growth regulators), and comparing predicted yields to historical or expected yields for specific fields.

While crop yield models have demonstrated adequate performances at the field scale, their applications across large areas face challenges due to the substantial data volume and computational processing costs [10]. To address them, statistical regression models such as multiple linear regression and machine learning techniques have been employed to integrate spectral indices from satellite data with climate variables at regional or national levels [10–12]. Table 1 presents various studies on cotton yield estimation utilizing statistical regression models or crop models combined with remote sensing across the regional and field scale under different platforms (satellite, airplane, unmanned aerial vehicle (UAV), and ground sensors) in diverse countries. Three of these studies developed regression models, including Random Forest, to forecast cotton yield at a regional scale in China [10], India [11] and Australia [12], using satellite-derived vegetation indices (VIs) and climate variables as covariates. The models incorporated precipitation, temperature, evapotranspiration, vapor pressure, soil moisture, and other climate variables. They utilized hundreds of data instances from thousands of hectares across different farm fields over multiple crop years for training and testing, resulting in root mean square errors (RMSE) for seed cotton yield of 157 kg ha$^{-1}$ [10], 375 kg ha$^{-1}$ [11], and 976 kg ha$^{-1}$ [12] in China, India, and Australia, respectively.

Simple linear regression models have also been developed to predict cotton yield using solely satellite-derived vegetation indices as independent variables [13–16], unmanned aerial vehicle (UAV) [17–20], airplane [21], and ground sensor [22]. While vegetation indices alone have limitations in capturing the complex relationships among production, plant physiology, climate, soil nutrition, soil water parameters, pest and disease infestation, and crop management characteristics [10], several studies have demonstrated their ability to predict crop yield with reasonable accuracy [7,23,24]. On average, the RMSEs for seed cotton yield for the linear regressions were comparable to those assessed by using machine learning techniques and crop model approaches (Table 1), indicating the potential advantages of linear regression models. These models operate with a smaller set of well-defined independent variables, require less data manipulation, and utilize smaller data instances compared to other approaches. Moreover, the greater mathematical explainability inherent in linear equations for specific satellite products, vegetation indices, and crop phenological stages allows for direct transferability and easy adaptation to other locations, facilitating clearer understandings and comparisons among different models.

The current study assesses the feasibility of employing simple linear models to correlate cotton yield with VIs derived from satellite data obtained from the Moderate-Resolution Imaging Spectroradiometer (MODIS) sensors aboard the Terra and Aqua satellites. The aim is to forecast in-season cotton yield within the Brazilian Cerrado region. Reflectance time series data from multiple MODIS spectral bands were extracted to assess 15-day interval averages spanning from sowing to harvest of widely recognized VIs. These VI averages, computed over various 15-day intervals, were accessed as predictors (independent variables) within linear regression models to estimate cotton yield.

**Table 1.** A review of previous studies for estimation of seed cotton yield using remote sensing.

| Reference | Country | Plot | Area ha | Model Aproach | RS Source | Regression Model | RMSE kg ha$^{-1}$ | R$^2$ |
|---|---|---|---|---|---|---|---|---|
| [10] | China | 355 | - | CV/RS | Modis/Sentinel | LSTM, SVM, RF | 375 | 0.65 |
| [25] | EUA | 12 | 150 | CM/RS | Spectroradiometer | | 468 | - |
| [13] | USA | 3 | 188 | RS | Modis/Landsat | LR | 673 | 0.52 |
| [14] | USA | - | - | RS | Modis | LR | - | 0.16 |
| [11] | India | - | - | CV, RS | Modis | RF | 157 | 0.69 |
| [12] | Australia | 253 | - | CV, RS | Landsat | RF | 976 | - |
| [17] | USA | 1 | 5 | RS | UAV | MLR | 261 | 0.87 |
| [26] | USA | 805 | 0.65 | RS | UAV | ANN, RF | - | 0.72 |
| [27] | USA | 1 | 57 | RS | Modis/Landsat | | 463 | 0.84 |
| [28] | USA | - | - | EM/RS | Sentinel | | - | - |
| [18] | USA | 2550 | 6 | RS | UAV | LR | 550 | 0.92 |
| [29] | Brazil | 1 | 90 | RS | Optical sensor | decision trees | - | 0.81 |
| [30] | USA | - | 73 | RS | Landsat | ANN | 375–470 | 0.71 |
| [31] | USA | 2 | 120 | RS | Landsat | exponential | 481 | 0.81 |
| [19] | Australia | 90 | 7 | RS | UAV | LR and quadradic | - | 0.75 |
| [15] | USA | 949 | - | RS | Modis | LR | - | 0.48 |
| [16] | USA | 24 | 0.2 | RS | NASA data | LR | - | 0.85 |
| [21] | USA | 48 | 1.5 | RS | Airborne | LR | - | 0.47 |
| [22] | USA | 44 | 5.3 | RS | Spectroradiometer | LR | - | 0.89 |

CV: Climate Variables, RS: Remote Sensing, CM: Crop Model, EM: Ecosystem Model, RS: Remote Sensing, UAV: Unmanned Aerial Vehicle, LSTM: Long Short-Term Memory, SVM: Support Vector Machine, RF: Random Forest, LR: Linear Regression, MLR: Multiple Linear Regression, ANN: Artificial Neural, Network, RMSE: Root Mean Square Error of seed cotton yield, R$^2$: Coefficient of determination.

## 2. Materials and Methods

### 2.1. Experimental Areas and Dataset

The regression models utilized data from 281 commercial cotton farm fields (hereinafter referred to as plots) across the states of Mato Grosso (125 plots), Goiás (83 plots), and Bahia (73 plots) during the growing seasons from 2016 to 2022. Data from 2016–2018 seasons were gathered from surveys conducted by the Mato Grosso Cotton Institute [32,33] and the Brazilian Agricultural Research Corporation [34], while data from the 2019 to 2022 seasons were obtained directly from cotton farmers. Figure 1 illustrates the spatial distribution of the 281 plots. Most plots in Mato Grosso, Goiás, and Bahia were situated at the western, southern, and western parts of the states, respectively.

The dataset contains information on seed cotton yield (kg ha$^{-1}$), plot area (ha), cultivar, plant line spacing (m), seed population (seed ha$^{-1}$), sowing date, and water management (irrigated vs. rainfed) for 281 plots. The average seed cotton yield was 4209 kg ha$^{-1}$, ranging from 694 kg ha$^{-1}$ to 7361 kg ha$^{-1}$. Low-yield plots occurred in rainfed plots located in Mato Grosso and Goiás during the 2016 and 2022 seasons, likely due to drought or irregular rainfall [35]. Conversely, most of the high-yielding plots were found in Bahia under central pivot irrigation and in Mato Grosso and Goiás during seasons with favorable rainfall distribution (2017 and 2018). Only 12% (34) of the plots were irrigated, all located in Bahia (21) and Goiás (13). The remaining 88% (247 plots) were rainfed, with plots in Mato Grosso (125), Bahia (52), and Goiás (70) relying solely on rainfall. Plant line spacing was predominantly 0.9 m, with an average seed linear density of 10 seed m$^{-1}$, resulting in an average population of 111,111 seed ha$^{-1}$. The average plot area was 188 ha and the average cotton cycle lasted 196 days. Cultivars FM975WS, FM944GL, FM913GLT, and FM940GLT (FiberMax, Basf, São Paulo, SP, Brazil), TMG42WS, TMG47B2RF, and TMG81WS (Tropical, Melhoramento e Genética, Londrina, PR, Brazil), DP1243B2RF and DP1536B2RF (Deltapine, Bayer, São Paulo, SP, Brazil), and IMA8405 (Instituto Matogrossense do Algodão, Cuiabá, MT, Brazil) were the dominant choices for cotton growers in this study, representing approximately 80% of all planted varieties.

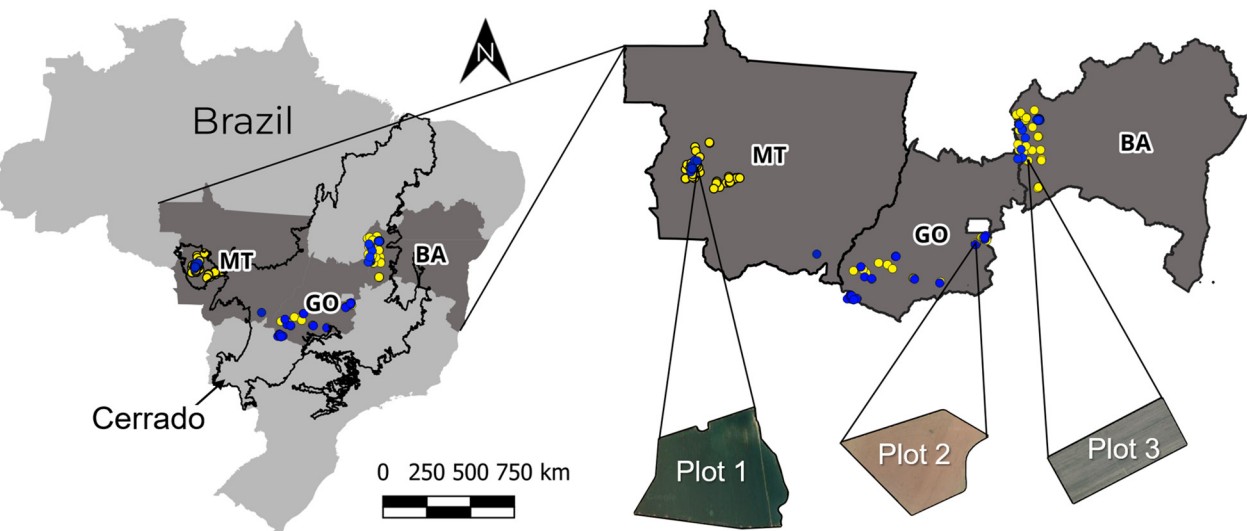

**Figure 1.** Spatial location of the 281 plots cultivated with cotton in the states of Mato Grosso (MT), Goiás (GO), and Bahia (BA) (typical Brazilian Cerrado biome) used to train (yellow dots) and test the models (blue dots) and three cotton plots zoomed (Plots 1, 2, and 3).

Daily accumulated precipitations and extreme air temperatures (maximums and minimums) data were obtained for each plot by mean of the Brazilian agrometeorological monitoring system (web application) AGRITEMPO [36], which collects data from conventional and automatic weather stations, by taking the closest station to each plot and downloading the daily climate data from 0 to 180 days after sowing (DAS). The data were then averaged for month 1 (0–30 DAS) to 6 (150–180 DAS) and then correlated to the observed yield only to evaluate the effects of the climate variables on yield and to help explain possible bias between observed and predicted yields.

### 2.2. Satellite Data Acquisition and Preprocessing

With geographic co-ordinates of all plots, contour maps (kml format) were generated using the Google Earth Pro (Google, Mountain View, CA, USA) and then converted to shapefile format using the QGIS software [37]. Spectral reflectance data and VIs were obtained as time series from the EOS-MODIS MCD43A4 V6.1 product [38]. This product provides daily surface reflectance data (NBAR) adjusted for Nadir bidirectional reflectance distribution function (BRDF) effects. It offers smoothed data with a 500 m spatial resolution. The product uses 16-day compositions with the best pixel selection (maximum value composite) to create daily time series. The product includes spectral bands for blue (0.459–0.479 µm), green (0.545–0.565 µm), red (0.62–0.67 µm), near infrared (NIR) (0.841–0.876 µm), short-wave infrared (SWIR1) (1.23–1.25 µm), SWIR2 (1.628–1.656 µm), and SWIR3 (2.105–2.155 µm). Google Earth Engine (GEE) platform [39], and its JavaScript code editor, was used to extract these time series. The GEE code selects pixels within each plot contour area, excluding those within 250 m of the boundary using a buffer command. This step ensured focus on the core area of each plot. Finally, the code calculated and exported average reflectance values for each band across the entire growing season (from sowing to harvest). The reflectance image and the time series graph were visually inspected before exporting the representative time series for each plot in comma-separated values (csv).

Spectral bands were then combined in an Excel®, (Tokyo, Japan) worksheet and the VIs were calculated, averaged in 15-day intervals from 0 to 240 DAS, and correlated to cotton yield for the each mutually exclusive interval. Reflectance data are plotted in a logarithmic scale for better comparisons among the spectral bands. In Table 2 is shown a list of the VIs and their definitions. A subset of plots (167) was randomly selected (balanced by yield) and used for generating the regression models (training set). The remaining dataset

(114 plots) was used for the independent model validation (test set), employing a well-known cross-validation method set as 60–40% (train–test). Linear regression were fitted for each combination of VI (Table 2) and time interval. Models' accuracies were measured by the Root Mean Square Error (RMSE) and determination coefficient ($R^2$) estimated from the test set. Linear regressions were performed, and the RMSE, $R^2$, and statistical significance (Analysis of Variance—ANOVA, New Providence, NJ, USA) of the fittings were calculated using a spreadsheet program (Microsoft® Excel® 2019, version 2403).

**Table 2.** Vegetation indices used in this study.

| Vegetation Indices | Formulation | Reference |
|---|---|---|
| Green Index (GI) | GI = G/R | [40] |
| Ratio Vegetation Index (RVI) | RVI = NIR/R | [41] |
| Chlorophyll Vegetation Index (CVI) | CVI = (NIR × R)/$G^2$ | [42] |
| Soil-Adjusted Vegetation Index (SAVI) | SAVI = 1.5 × (NIR − R)/(0.5 × NIR + R) | [43] |
| Chlorophyll index—green (CIG) | CIG = (NIR/G) − 1 | [40] |
| Triangular Chlorophyll Absorption Ratio Index (TVI) | TVI = 60 × NIR − G − 100 × (R – G) | [44] |
| Green NDVI (GNDVI) | GNDVI = (NIR − G)/(NIR + G) | [45] |
| Enhanced Vegetation Index (EVI) | EVI = 2.5 × (NIR − R)/(1 + NIR + 6 × R − 7.5 × B) | [46] |
| Normalized Differential Vegetation Index (NDVI) | NDVI = (NIR − R)/(NIR + R) | [47] |

G: green, R: red, NIR: near infrared, B: blue.

## 3. Results and Discussion

Examples of MODIS images (NIR band) and time series of all spectral reflectance bands are shown in Figure 2 for a plot with very low cotton yield (1665 kg ha$^{-1}$ in season 2016; Figure 2a, left) and high yield (4965 kg ha$^{-1}$ in season 2017; Figure 2a, right). The original smoothed daily time series are shown in Figure 2b and 15-day averaged time series in Figure 2c. Reflectance data are plotted in a logarithmic scale for better comparisons among the spectral bands. In general, lower reflectance values (greater absorption of electromagnetic radiation) are observed for the blue, red, and green bands due to the absorption of these light wavelengths by leaves through photosynthesis. Higher reflectance in this range can be associated to plant stresses caused by biotic and/or abiotic factors. The spectral range between 0.4 and 0.7 μm (visible region) is the photosynthetically active radiation (PAR) that is the most efficient portion of the electromagnetic spectrum used by the plant for photosynthesis. Healthy plants present high light absorption in the visible range (0.4–0.7 μm) mainly by the leaf pigments like chlorophyll and carotenoids, relatively high reflectance in the NIR range (0.7–1.1 μm) due to leaf and cell structure scattering effects, and relatively low reflectance in the SWIR range (1.1–2.5 μm) due to water and chemicals into the leaf structure [48,49]. For an actual instance, the low cotton yield (left graph in Figure 2) was mainly caused by low precipitation (water stress) in the 2016 season [35], which is noted by slight increases in red and blue reflectance (lower absorbance by cotton plants), especially between 70 and 170 DAS, and decreasing NIR reflectance, as compared to the high-yield plot (right).

Figure 3a shows the average time series of the seven MODIS spectral bands for five classes of cotton yield (lower than 3, 3–3.75, 3.75–4.5, 4.5–5.25, and greater than 5.25 ton ha$^{-1}$). As expected, the reflectance of NIR and SWIR1 bands increased with higher cotton yield. Conversely, reflectance in the red, blue, SWIR2, and SWIR3 decreases as yield increases, particularly from around 60–80 DAS until harvest. This trend is confirmed by the correlation coefficients (r) between cotton yield and reflectance displayed in Figure 3b. Notably, the green band reflectance, while higher than red and blue as expected, exhibits no clear visual relationship with yield in Figure 3a. Figure 3b highlights the strongest correlations between spectral reflectance bands and cotton yield. Positive correlations are observed for NIR and SWIR1 bands, while negative correlations are found for SWIR3, red, SWIR2, and blue bands. These findings, obtained using broadband satellite sensors, are consistent with studies employing narrow-band ground measurements with spectrora-

diometers. For example, Thenkabail et al. [50] reported a negative correlation between cotton biophysical parameters (leaf area index, wet biomass, plant height, and crop yield) for reflectance in the visible light range (350 nm to 700 nm) and positive correlations in the near-infrared to short-wave infrared range (730 nm to 1050 nm).

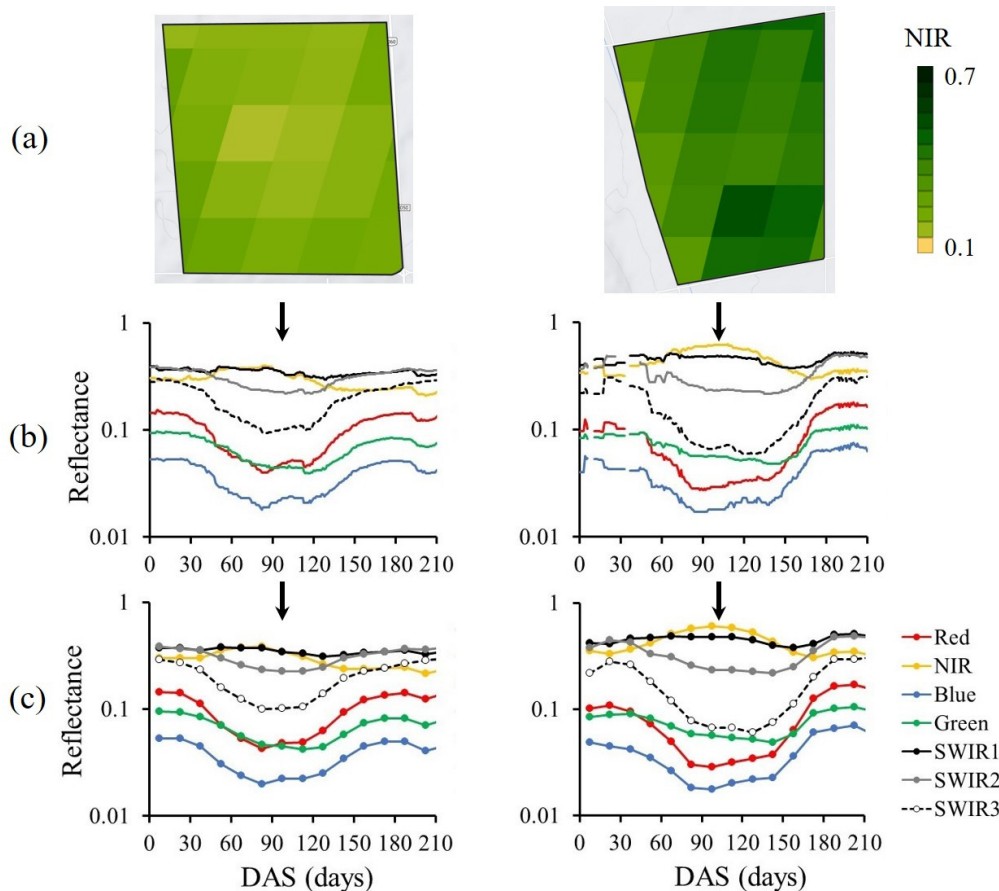

**Figure 2.** MODIS product MCD43A4 V6.1: (**a**) NIR images obtained for two commercial cotton production plots; (**b**) original extracted time series (daily data) for red, NIR, blue, green, and SWIR spectral bands; (**c**) time series in 15-day interval averages. Plot on left presented low cotton yield on right high cotton yield.

Cotton yield prediction for each 15-day interval during the growing season was attempted using simple linear regression models. The nine vegetation indices listed in Table 2 were used as independent variables and the training dataset (167 plots) was employed for model fitting. Figure 4b displays the determination coefficients ($R^2$) achieved for these VIs across DAS intervals. The highest $R^2$ values were obtained for the TVI (0.69), EVI (0.68), SAVI (0.67), and NDVI (0.61) during intervals between 120 and 165 DAS. This period corresponds to the cotton late-season stage, encompassing boll opening and defoliation. Figure 4a presents the average time series of these VIs for different cotton yield classes. A general trend of saturation was observed for all VIs at the peak in the time series for the highest yield classes. This effect was most pronounced for NDVI, which is consistent with previous studies reporting NDVI saturation at high levels of green biomass or leaf area index [51,52].

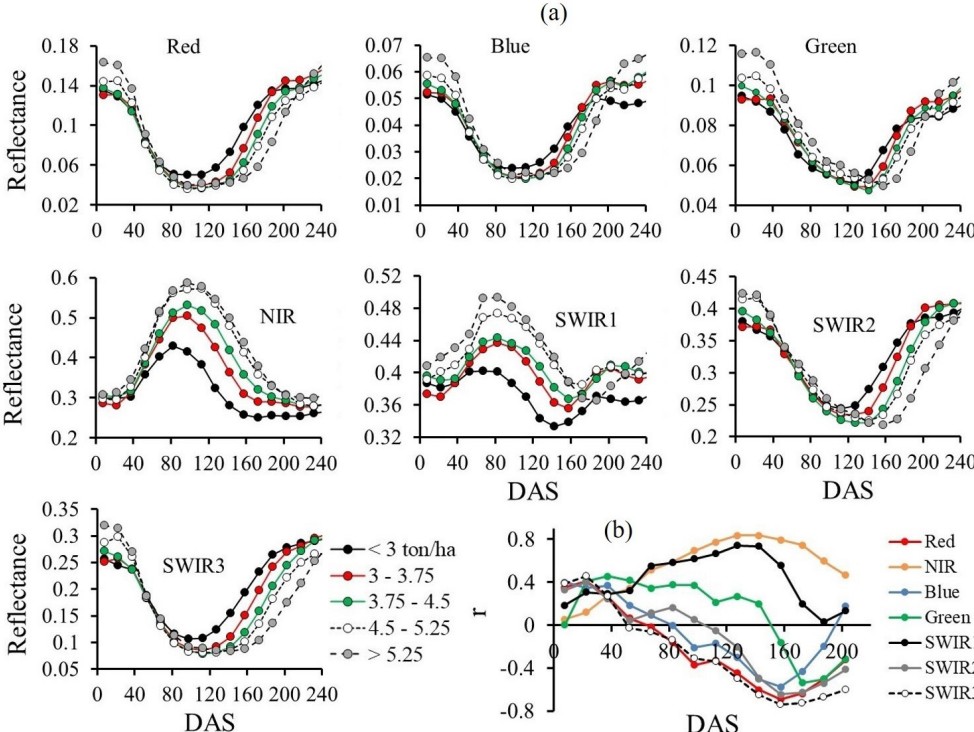

**Figure 3.** Average time series of reflectance for seven MODIS spectral bands for five cotton yield classes (**a**) and the linear correlation coefficients (r) between cotton yield and reflectance for the different 15-day days after sowing (DAS) intervals in the training dataset (167 plots) (**b**).

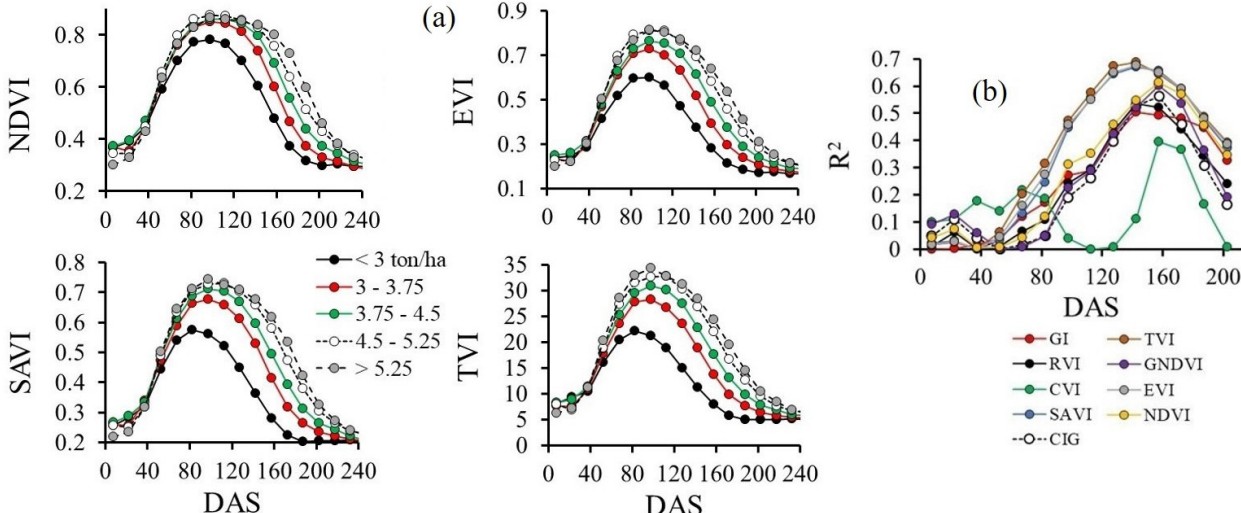

**Figure 4.** Time series of NDVI, EVI, SAVI, and TVI from MODIS sensor for five cotton yield classes (**a**) and determination coefficients ($R^2$) of linear correlations between cotton yield and 9 vegetation indices (Table 2) (**b**) for the different 15-day days after sowing (DAS) classes for the training dataset (167 plots).

The results suggest that TVI, EVI, and SAVI hold promise for in-season cotton yield estimation within the training dataset, as they yielded the highest determination coefficients for yield prediction. Table 3 presents the linear regression models for the four VIs with the highest $R^2$ and statistically significant overall fit (analysis of variance, ANOVA, *p*-value < 0.05) of the adjusted models for different 15-day intervals, ranging from 75–90 to 180–195 DAS. The table also includes models for the average VI from 75 DAS to harvest (75–195 DAS) and the peak of the observed VI values throughout the season. Figure 5

illustrates the relationships between cotton yield and TVI for different DAS intervals. While the highest $R^2$ (0.73) was achieved for the 75–195 DAS interval for TVI (Figure 5) and the other VIs (Table 3), these equations would only be practical at the very end of the season since they require data accumulation until 195 DAS for averaging. For in-season forecasting, models derived from VI data between 105–120 DAS and 165–180 DAS yielded the most promising results. However, intervals like 150–165 and 165–180 DAS are too close to harvest and may not provide sufficient lead time for farmers to implement management interventions effectively. Peak VI values have been explored in various studies for in-season yield estimation [14,53]. In this study, the peak models exhibited lower $R^2$ values compared to those obtained between 105–120 and 165–180 DAS (Table 3) but remained higher than those for 90–105 DAS interval. This suggests potential for earlier in-season forecasts considering that peak VI values for TVI, EVI, SAVI, and NDVI typically occur around 80 to 100 DAS, as seen in Figure 4.

**Table 3.** Linear regression models for predicting cotton yield from averaged 15-day TVI, EVI, SAVI, and NDVI vegetation indices, averaged 75–195 DAS and peak (maximum value in the time series), using the training dataset.

| DAS | Linear Model (TVI) | $R^2$ | RMSE | *p*-Value | DAS | Linear Model (EVI) | $R^2$ | RMSE | *p*-Value |
|---|---|---|---|---|---|---|---|---|---|
| | | | kg ha$^{-1}$ | ($\alpha$ = 0.05) | | | | kg ha$^{-1}$ | ($\alpha$ = 0.05) |
| 75–90 | Y = 111.96 TVI + 794.22 | 0.31 | 1088 | $3.3 \times 10^{-15}$ | 75–90 | Y = 5133.1 EVI + 315.9 | 0.28 | 1119 | $4.1 \times 10^{-13}$ |
| 90–105 | Y = 139.78 TVI − 90.918 | 0.47 | 953 | $9.5 \times 10^{-25}$ | 90–105 | Y = 6863.1 EVI − 1066.8 | 0.46 | 966 | $9.0 \times 10^{-24}$ |
| 105–120 | Y = 152.11 TVI − 269.52 | 0.58 | 856 | $1.7 \times 10^{-32}$ | 105–120 | Y = 7316.1 EVI − 1249.5 | 0.55 | 879 | $1.4 \times 10^{-30}$ |
| 120–135 | Y = 152.87TVI + 166.22 | 0.67 | 752 | $7.6 \times 10^{-42}$ | 120–135 | Y = 7013.0 EVI − 608.5 | 0.65 | 778 | $2.3 \times 10^{-39}$ |
| 135–150 | Y = 148.0 TVI + 878.89 | 0.69 | 735 | $1.8 \times 10^{-43}$ | 135–150 | Y = 6484.9 EVI + 292.1 | 0.68 | 778 | $2.7 \times 10^{-42}$ |
| 150–165 | Y = 147.88 TVI + 1553.7 | 0.64 | 783 | $6.5 \times 10^{-39}$ | 150–165 | Y = 6170.2 EVI + 1127.9 | 0.65 | 776 | $1.5 \times 10^{-39}$ |
| 165–180 | Y = 162.65 TVI + 1957.8 | 0.59 | 843 | $1.4 \times 10^{-33}$ | 165–180 | Y = 6456.5 EVI + 1636.5 | 0.59 | 842 | $1.1 \times 10^{-33}$ |
| 180–195 | Y = 195.06 TVI + 2130.5 | 0.49 | 941 | $1.1 \times 10^{-25}$ | 180–195 | Y = 7602.2 EVI + 1801.6 | 0.48 | 947 | $3.2 \times 10^{-25}$ |
| 75–195 | Y = 196.28 TVI − 189.77 | 0.72 | 690 | $5.9 \times 10^{-48}$ | 75–195 | Y = 8891.9 EVI − 1026.2 | 0.73 | 688 | $3.9 \times 10^{-48}$ |
| Peak | Y = 150.52 TVI − 634.87 | 0.53 | 900 | $7.4 \times 10^{-29}$ | Peak | Y = 7813.9 EVI − 1996.1 | 0.53 | 901 | $8.4 \times 10^{-29}$ |

| DAS | Linear Model (SAVI) | $R^2$ | RMSE | *p*-value | DAS | Linear Model (NDVI) | $R^2$ | RMSE | *p*-value |
|---|---|---|---|---|---|---|---|---|---|
| | | | kg ha$^{-1}$ | ($\alpha$ = 0.05) | | | | kg ha$^{-1}$ | ($\alpha$ = 0.05) |
| 75–90 | Y = 6804.1 SAVI − 543.3 | 0.25 | 1140 | $8.2 \times 10^{-12}$ | 75–90 | Y = 6111.6 NDVI − 1057.4 | 0.12 | 1232 | $4.5 \times 10^{-6}$ |
| 90–105 | Y = 9495.4 SAVI − 2479.4 | 0.45 | 977 | $5.8 \times 10^{-23}$ | 90–105 | Y = 11,059 NDVI − 5334.6 | 0.31 | 1089 | $4.3 \times 10^{-15}$ |
| 105–120 | Y = 10,008 SAVI − 2696 | 0.55 | 1132 | $6.1 \times 10^{-30}$ | 105–120 | Y = 11,093 NDVI − 5287.1 | 0.35 | 1058 | $3.3 \times 10^{-17}$ |
| 120–135 | Y = 9164.3 SAVI − 1712.5 | 0.64 | 784 | $8.2 \times 10^{-39}$ | 120–135 | Y = 9500.8 NDVI − 3674.2 | 0.46 | 966 | $8.7 \times 10^{-24}$ |
| 135–150 | Y = 7937.6 SAVI − 415.88 | 0.67 | 753 | $9.7 \times 10^{-42}$ | 135–150 | Y = 7473.6 NDVI − 1618 | 0.55 | 884 | $3.6 \times 10^{-30}$ |
| 150–165 | Y = 7729.7 SAVI + 624.53 | 0.66 | 809 | $5.6 \times 10^{-40}$ | 150–165 | Y = 6243.6 NDVI − 80.36 | 0.61 | 816 | $6.0 \times 10^{-36}$ |
| 165–180 | Y = 7244.4 SAVI + 1251.7 | 0.59 | 840 | $7.9 \times 10^{-34}$ | 165–180 | Y = 5550.8 NDVI + 1002 | 0.57 | 861 | $4.8 \times 10^{-32}$ |
| 180–195 | Y = 8316.9 SAVI + 1422.3 | 0.47 | 952 | $7.5 \times 10^{-25}$ | 180–195 | Y = 5922.8 NDVI + 1386.4 | 0.46 | 970 | $1.6 \times 10^{-23}$ |
| 75–195 | Y = 11,152 SAVI − 2066.5 | 0.73 | 688 | $1.9 \times 10^{-48}$ | 75–195 | Y = 11,230 NDVI + 4003.9 | 0.67 | 752 | $8.1 \times 10^{-42}$ |
| Peak | Y = 11,044.7 SAVI − 3770 | 0.51 | 919 | $2.3 \times 10^{-27}$ | Peak | Y = 15,470 NDVI − 9131.6 | 0.39 | 1029 | $3.2 \times 10^{-19}$ |

Y: seed cotton yield (kg ha$^{-1}$); Peak: maximum value of the VI in the time series; *p*-value determined by ANOVA.

Model estimates for cotton yields using TVI equations (Table 3) are compared to observed yields alongside their respective RMSE in Figure 6. Figure 7 illustrates variations in RMSE (Figure 7a) and $R^2$ (Figure 7b) across different DAS intervals for the four best VIs. The lowest RMSE and highest $R^2$ were achieved by EVI and TVI between 90–105 and 135–150 DAS (four DAS intervals), enabling in-season yield forecasting with RMSE of approximately 750 kg ha$^{-1}$ (Figure 7a). Averaged VIs for the period 75–195 DAS and peak models displayed comparable RMSE values to the best models (EVI, TVI, and SAVI) for 15-day intervals (90–105 to 135–150 DAS) (Figure 7c). However, for EVI, the peak model exhibited the lowest RMSE (726 kg ha$^{-1}$). Figure 7d presents the average yield predictions for the 114 testing plots at 105–120 DAS, 75–195 DAS, and peaks. The average observed yield for the 114 plots stood at 4372 kg ha$^{-1}$ (represented by the dotted horizontal line in Figure 7d), with most linear models tending to overestimate yield by approximately 160 kg ha$^{-1}$ on average. Overall, there were tendencies of overestimation for lower-yield plots and underestimation for higher-yield plots, as evidenced in Figure 6 for TVI. However, the 75–195 DAS model appears to reduce such bias. Although NDVI provided a closer average yield forecast for the 114 plots compared to the measured average yield (Figure 7d),

its RMSE was significantly higher than the other VIs (EVI, TVI, and SAVI) for all intervals between 75 and 165 DAS. Therefore, the best performances were achieved with EVI and TVI for the 105–120, 120–35, and 135–150 DAS intervals. If an earlier prediction is required, the 90–105 DAS or peak models are preferred, as the RMSE for 75–90 DAS and earlier DAS intervals were excessively high, and the $R^2$ values were insignificant.

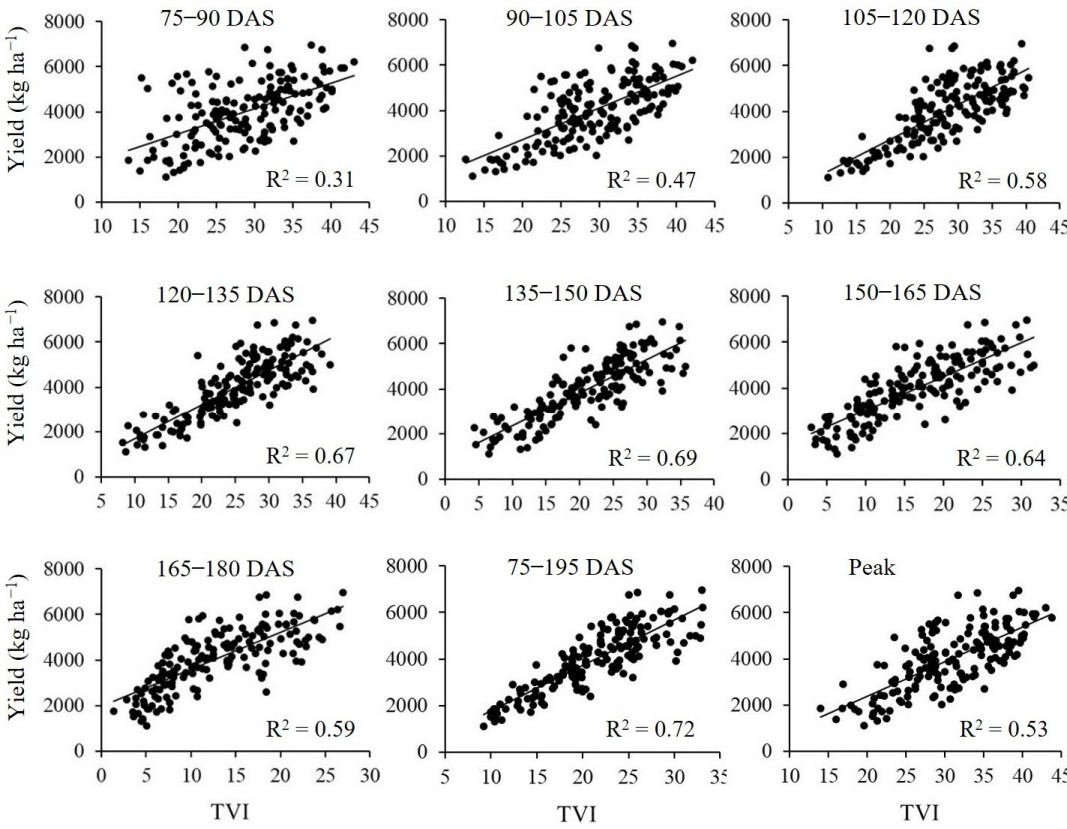

**Figure 5.** Linear correlations between TVI and seed cotton for different 15-day DAS for the training dataset, from 75–90 to 180–195 DAS, 75–195 DAS (averaged period) and the time series peak value. Linear equations shown in Table 3.

The observed trend of overestimation for low-yield plots and underestimation for high yields (Figure 6) can be further examined by correlating the residual error (yield$_{estimated}$ − yield$_{observed}$; estimated by the peak model with TVI) for the observed yield of each plot (Figure 8a). In this analysis, negative residual errors (indicating underestimation) are evident for yields higher than approximately 4000 kg ha$^{-1}$, while positive residual errors (indicating overestimation) are observed for lower yield values. Several factors may contribute and explain the variation in low and high yield plots, such as climate conditions, soil type, soil fertility, diseases, cultivars, and others. However, in rainfed cotton production, one of the primary driven factors affecting yield has consistently been climatic condition, particularly variations in precipitation and temperature throughout the season. Consequently, the choice of sowing date and cycle duration, determined based on the recommended cultivar and local climatic patterns, is influenced by water demands and temperature fluctuations during each stage of cotton development, ultimately impacting the expected cotton yield. For the testing dataset, the earliest sowing date was 19 November. Therefore, observed yield was correlated with sowing dates after 1 November as a proxy for sowing data, independent of the year (Figure 8b), revealing a negative influence on yield. Additionally, the cotton cycle duration, primarily dictated by the cultivar used, exhibited a positive correlation with yield (Figure 8c). These two parameters are intrinsically linked to

climatic conditions and both significantly influence observed yields, with a discernible trend of increasing yield associated with earlier sowing dates and longer cotton cycle durations.

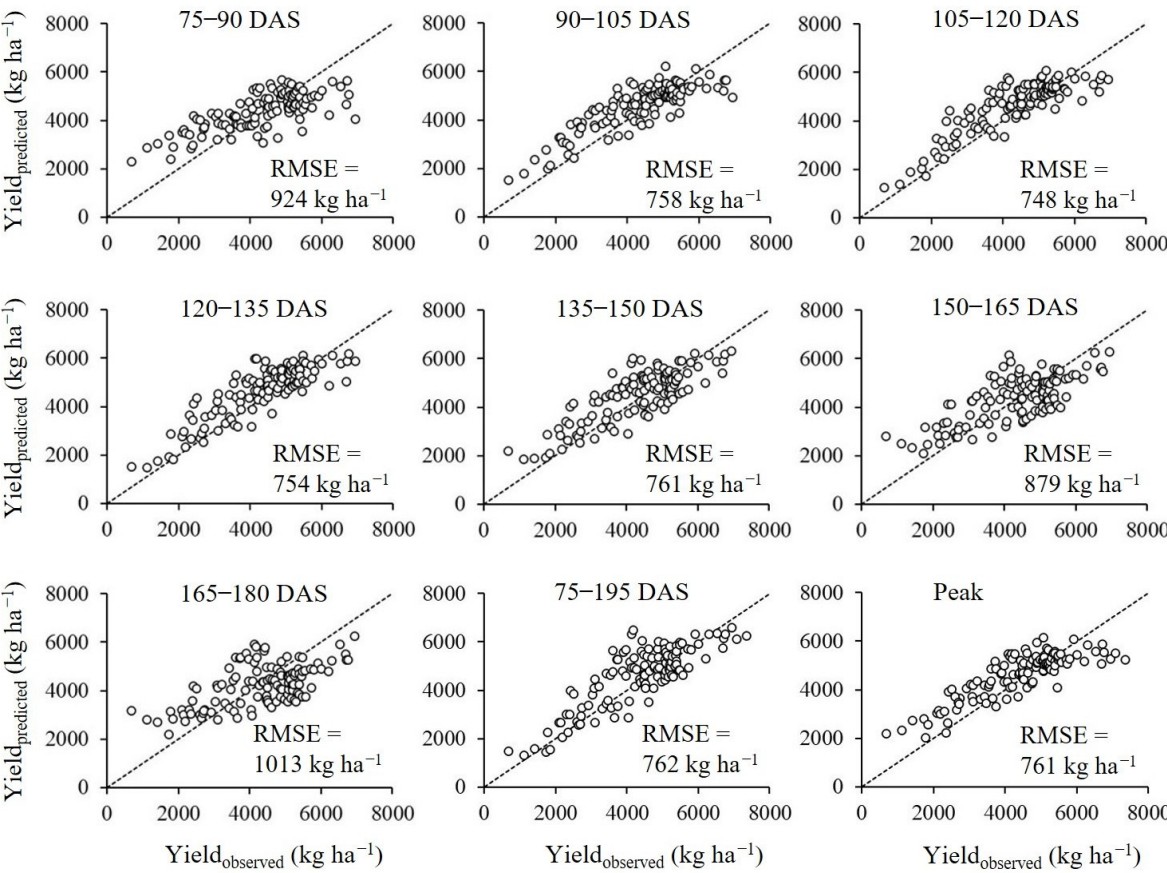

**Figure 6.** Comparisons between observed and predicted seed cotton yield for the testing dataset (114 plots) for different 15-day DAS (75–90 to 180–195 DAS), 75–195 DAS (averaged period), the time series peak value, and their respective root mean square errors (RMSE), obtained using TVI linear regression models (Table 3).

The most significant correlations between monthly accumulated precipitation and yield were observed in the third and fourth months after sowing (r = 0.59 for 60–90 DAS and 0.39 for 90–120 DAS; Figure 8d), corresponding to the mid-season of cotton growth, from canopy closure until flowering and boll development stages. In the first month, rainfall demonstrated an adverse effect on yield (r = −0.29), as high-intensity rainfall during the early season can potentially damage the germination process. Monthly averaged maximum and minimum temperatures, along with their differences ($T_{max} - T_{min}$) also exerted an influence on yield, particularly after the third month (r = −0.49).

Several studies, such as those listed in Table 1, have investigated the optimal timing for acquiring satellite and UAV images or ground measurements for cotton yield predictions. Some studies have relied on single-date images, while others have assessed images at specific phenological stages or analyzed complete time series data. Table 4 provides a summary of the optimal periods, expressed in DAS, as inferred from the literature cited in Table 1. The most favorable correlations between cotton yield and the VIs have been reported across various stages, ranging from the flowering and boll development period [12,18,22,26–31] to boll opening, maturation, and defoliant application [10,15–17,19,21,27,30]. Notably, these findings align with the results obtained in the present study. Specifically, the study conducted by Lang et al. [10] in Xinjiang Province, China, spanning from 2012 to 2019 and encompassing 355 plots, yielded results strikingly similar to ours, with the lowest

RMSE and highest R$^2$ observed during the fourth and fifth months after sowing (90–120 to 120–150 DAS).

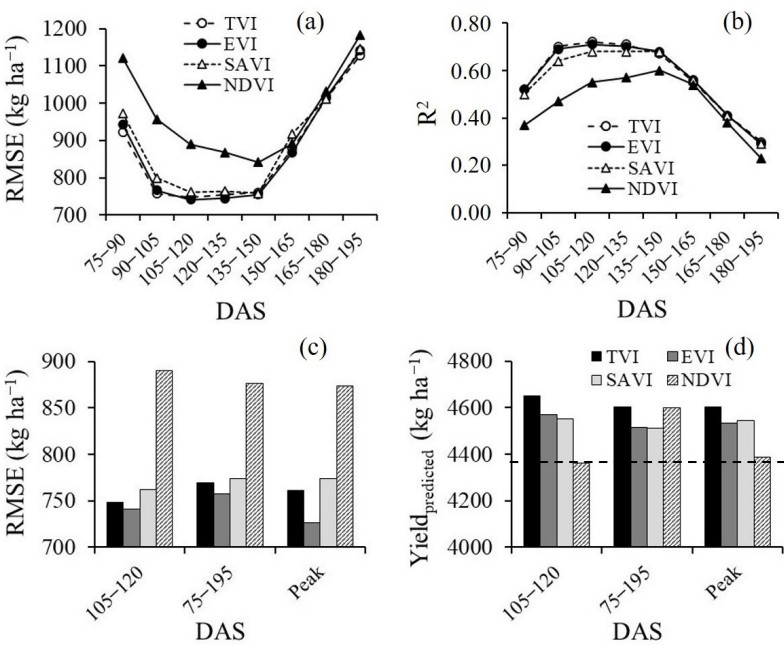

**Figure 7.** Root mean square error (RMSE) (**a**) and linear determination coefficients (R$^2$) (**b**) between predicted and observed yield for the 114 plots of the testing dataset for different 15-day DAS; comparison of RMSE (**c**) and predicted yield (**d**) for 105–120 DAS, 75–195 DAS, and peak equations for TVI, EVI, SAVI, and NDVI. Dotted horizontal line in (**d**) represents the average observed yield for the 114 plots.

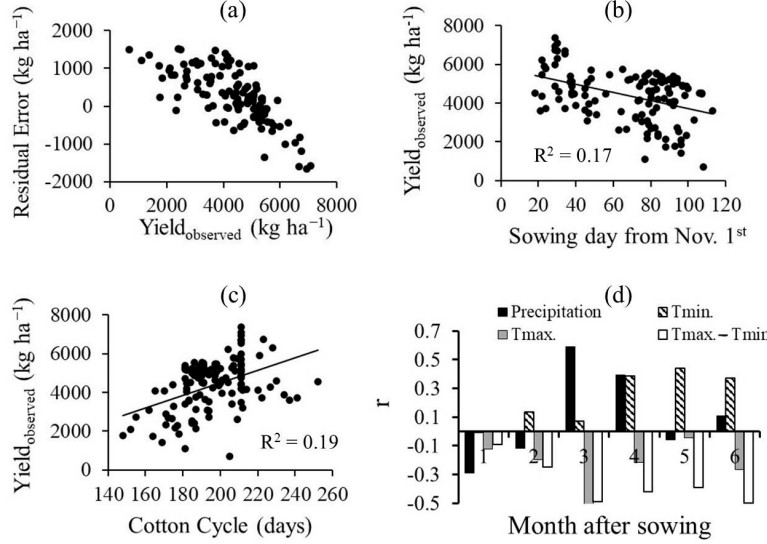

**Figure 8.** Relationships between the residual error (Yield$_{estimated}$ − Yield$_{observed}$; estimated by TVI peak model) and observed cotton yield (**a**); observed yield and sawing day from 1 November (**b**); observed yield and cotton cycle (**c**); and correlation coefficients (r) between observed yield and monthly accumulated precipitation, monthly averaged minimum (Tmin) and maximum (Tmax) temperatures, and Tmax − Tmin (**d**) for month 1 to 6 after sowing.

**Table 4.** Optimal periods for remote sensing (RS) image or data acquisition, expressed in days after sowing (DAS), considering best correlations between yield and VIs from different studies.

| Reference | RS | DAS | Reference | RS | DAS |
|-----------|----|----|-----------|----|----|
| [19] | UAV | 150–170 | [27] | Satellite | 100 |
| [30] | Satellite | 90–160 | [26] | UAV | 90–100 |
| [10] | Satellite | 90–150 | [12] | Satellite | 90 * |
| [15] | Satellite | 120 | [31] | Satellite | 80–90 |
| [21] | Airborne | 115 * | [18] | UAV | 80 * |
| [17] | UAV | 90–120 | [29] | Active Sensor | 60–100 |
| [16] | Satellite | 105 | [22] | Spectroradiometer | 75 |

* single date applied.

The most effective forecasting models for the mid to late-season period (90 to 150 DAS) on a regional scale can serve as valuable advance information for various stakeholders, including commodity traders, policymakers, governments, and, in some instances, farmers. This information can aid farmers in logistical planning and preparing for the next crop season, such as determining fertilizer needs based on the nutrient exportation from the previous crop. However, for certain in-season interventions like topdressing fertilizer applications, some forecasting models may not be practical. This is because most top-dressing fertilizers (both macro and micronutrients) are typically applied before 100 DAS. Nonetheless, earlier prediction using peak models still offers a window of opportunity for topdressing fertilizer intervention. For instance, by around 80 DAS, cotton plants have absorbed 45%, 50%, and 80% of sulfur, potassium, and nitrogen, respectively [54].

## 4. Conclusions

The forecasting methodology, employing simple regression models and time series intervals for cotton yield prediction using the MODIS product MCD43A4 V6.1 allowed in-season prediction with an RMSE of approximately 750 kg ha$^{-1}$ at a regional scale across three cotton-producing states in the Brazilian Cerrado. This straightforward approach offers ease of application and can be readily utilized for predicting seed cotton yield at farm, region, and national levels. One limitation of the approach is the coarse spatial resolution of the MODIS product, which restricts its applicability primarily to large commercial plots, characteristic of crop production systems in the Brazilian Cerrado.

Among the nine VIs evaluated, EVI and TVI emerged as the most effective individual predictors. However, it is important to note that accuracies, as assessed by mean of RMSEs, were notably low up to 75 DAS, presenting a limitation to the application of this approach for estimating cotton yield during the initial stages of cotton development. The most reliable in-season predictions were achieved through 15-day intervals ranging from 90–105 to 135–150 DAS, corresponding to the mid to late stages of cotton development (including boll development, open boll, and fiber maturation). For earlier stages (below 90–105 DAS), the most accurate forecasts were obtained from the model fitted for peaks using EVI and TVI, typically occurring around 80–90 DAS.

Future validation experiments should focus on assessing the accuracies and practical utility of various models in upcoming cotton seasons across different sub-regions. This evaluation should include their ability to predict cotton net production at various scales, ranging from individual farms to states, regions, and even at the national level. When considering the applicability of models at the farm level for specific management interventions, such as within-season topdressing fertilizer application or planning for the next crop based on exported nutrients, caution is warranted. Additional validations or complimentary approaches may be necessary to ensure the reliability and effectiveness of these models in practical agricultural decision making.

The proposed approach can be extended to corn, which is also commonly grown as a second harvest in the Brazilian Cerrado. Additionally, it could be adapted for soybean, the

primary first crop cultivated in Brazil, despite facing challenges such as increased cloud interference in satellite images due to its cultivation during the rainy season.

**Author Contributions:** Conceptualization, C.M.P.V. and F.S.d.S.; Data curation, R.G., J.L.B., M.d.S., F.J.P. and S.d.C.; Formal analysis, D.A.B.d.S., C.M.P.V. and F.S.d.S.; Investigation, D.A.B.d.S., C.M.P.V. and F.S.d.S.; Methodology, D.A.B.d.S., C.M.P.V., F.S.d.S., E.J.F. and J.C.F.; Resources, R.G., J.L.B., M.d.S., F.J.P. and S.d.C.; Software, F.S.d.S. and E.A.S.; Supervision, C.M.P.V.; Validation, D.A.B.d.S. and C.M.P.V.; Writing—original draft, D.A.B.d.S. and C.M.P.V.; Writing—review and editing, C.M.P.V., E.J.F., E.A.S., J.C.F., R.G., J.L.B., M.d.S. and F.J.P. All authors have read and agreed to the published version of the manuscript.

**Funding:** The authors acknowledge the financial support from the Brazilian Agricultural Research Corporation (EMBRAPA) and the Mato Grosso Cotton Institute (IMAmt) through a joint co-operation project (30.21.90.03) and thank the farmers in Mato Grosso, Bahia, and Goiás who voluntarily provided access to their commercial plots data.

**Data Availability Statement:** The data will be made available upon request to the corresponding author.

**Conflicts of Interest:** The authors declare no conflicts of interest.

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
