# Peer review of "Estimating Cotton Yield in the Brazilian Cerrado Using Linear Regression Models from MODIS Vegetation Index Time Series"

_agriengineering, doi:10.3390/agriengineering6020054_

Round 1

Reviewer 1 Report

Comments and Suggestions for Authors

Well done paper; could use a bit more emphasis on best uses and potential impact of adoption.

Author Response

Reviewer #1:

1.1. Comments and Suggestions for Authors: Well done paper; could use a bit more emphasis on best uses and potential impact of adoption.

Answer: We believe we have outlined the potential impacts of adopting the proposed procedure at farm, state, regional and national scale in the Discussion section (Page 12, Lines 343-353). In the Conclusion section (Page 13, Lines 373-378), we also emphasize the necessity for additional studies and validations across various scales to enable broader adoption.

Reviewer 2 Report

Comments and Suggestions for Authors

Regional-scale cotton yield estimate in the Brazilian Cerrado using simple linear regression models with time-series of vegetation indices from EOS-MODIS satellite data

General comments:

·      The work is novel, relevant and pertinent. In general, the work is well developed.

·      The document has some points that would help improve it.

·      Reduce the title and make it more general to make it more attractive.

·      Keywords should not be contained in the title.

·      Generally, result and discussion are presented separately. It is advisable to improve the discussion with more literature.

·      Consider modifying the conclusions and making them more concrete.

Abstract:

·      It is well structured.

·       

Introduction:

·      Include information on the economic impacts of predicting returns in the sector.

·      t is not common to present a table in the introduction. This information should be used more for discussion of the results.

·      Line 104-110. They look like they should be part of Materials and Methods.

Materials and Methods:

·      Line 175-177. The program or programs with which the regressions were carried out are not specified.

·      It is not specified whether the data meets the assumptions for making regressions: linearity, homoscedasticity, normality and independence. It should be included as part of Materials and Methods and Results.

Results and discussion:

·      In general there is little discussion with other authors, it is recommended to strengthen it. There is much recent literature related to the topic.

·      Consider the possibility of separating discussion results.

·      Line 183-186. It looks like it's part of Materials and Methods.

·      Line 228-230. It looks like it's part of Materials and Methods.

Conclusions:

·      Line 358-362. It seems like part of the discussion of results.

·      I suggest restructuring the conclusions. Conclude based on the objective and in a more concrete way.

Author Response

Reviewer #2:

2.1 The work is novel, relevant and pertinent.

Answer: No answer is needed.

2.2 In general, the work is well developed. The document has some points that would help improve it.

Answer: No answer is needed.

2.3 Reduce the title and make it more general to make it more attractive.

Answer: Taking into account the relevant recommendation, we changed the title to: “Estimating cotton yield in the Brazilian Cerrado using linear regression models from MODIS vegetation index time series”

2.4 Keywords should not be contained in the title.

Answer: We also changed the keywords to: remote sensing, cotton production forecast, Brazil

2.5 Generally, result and discussion are presented separately. It is advisable to improve the discussion with more literature.

Answer: AgriEngineering has accepted free-format submissions; the agglutination of Results and Discussion was a choice made by our research team based on usual practice in other journals. We carried out a new analysis of the referential literary scope provided in “Results and Discussion” section. We agree on the possibility of expanding discussions with more literature. However, as a result of our analysis, we understand that there are substantial references (21 in total) for guiding discussions. We understand and respect the recommendation, but it is the research team's desire to keep the section as it is.

2.6 Consider modifying the conclusions and making them more concrete.

Answer: We find the Conclusion section to be concise, comprising essentially two paragraphs. The first paragraph outlines the accuracies of the best models, while the second highlights the optimal vegetation indexes and forecast time periods. Furthermore, two additional paragraphs suggest future studies to independently validate the proposed models at different levels - farm, state, and country - and to explore the potential application of the proposed procedure to other crops in the Brazilian Cerrado.

2.7 Abstract: It is well structured.

Answer: No answer is needed.

2.8 Introduction: Include information on the economic impacts of predicting returns in the sector.

Answer: Unfortunately, economic analysis and/or econometric simulations for estimating contributions of yield forecasts are still scarce in the scientific literature, especially for Brazilian cotton producing regions. There will certainly be relevant impacts, but their econometric estimates will only be possible after the effective introduction of prediction models, such as those presented in this article. We agree information on economic impacts are relevant, but they are beyond the scope of this research.

2.9 Introduction: It is not common to present a table in the introduction. This information should be used more for discussion of the results.

Answer: Indeed, our research team shares the same vision as the reviewer. On the other hand, the exception for that is justified by the power to elucidate and easily compare multiple numerical and categorical data to justify and support the research. Additionally, we checked recent publications from the journal and found that this practice has been common and acceptable (e.g., Ennatiqi et al., 2024, 6, 724-753; Chaomuang et al., 2023, 5, 1865-1878; Al Kindi et al., 2023, 5, 2349-2365).

2.10 Introduction: Line 104-110. They look like they should be part of Materials and Methods.

Answer: We have excluded part of this paragraph, since this information is already in the Material and Methods section.

2.11 Materials and Methods: Line 175-177. The program or programs with which the regressions were carried out are not specified.

Answer: We really noticed the absence of the note... It has been included.

“Linear regressions were performed and the root mean squared error (RMSE) and coefficient of determination (R²) were calculated using a spreadsheet program (Microsoft Excel ® ).”.

2.12 Materials and Methods: It is not specified whether the data meets the assumptions for making regressions: linearity, homoscedasticity, normality and independence. It should be included as part of Materials and Methods and Results.

Answer: Indeed, the analysis of the linear models requires ensuring the assumptions and getting significance for model parameters (analysis of variance). We appreciate the note on an important aspect. We have included the ANalysis of Variance (ANOVA) in Table 3 to highlight the significance of the adjusted model (p-value <0.001). It is also essential to highlight that our approach was evaluated with a robust cross-validation method (train: 60% /  test: 40%), with relevant results. In general, when a model does not represent the implicit phenomenon, validation results show high errors disqualifying it for the task, which was not the case for the best results we showed in the article.

2.13 Results and discussion: In general, there is little discussion with other authors, it is recommended to strengthen it. There is much recent literature related to the topic.

Answer: We agree on the possibility of expanding discussions with more literature. However, as a result of revision, we are convinced that there are substantial references (21 in total) for guiding discussions. We understand and respect the recommendation, but it is the research team's desire to keep the section as it is”. See the complete addresses for that in note 2.5.

2.14 Results and discussion: Consider the possibility of separating discussion results.

Answer: AgriEngineering has accepted free-format submissions; the agglutination of Results and Discussion was a choice made by our research team based on usual practice in other journals.”. See the complete addresses for that in note 2.5.

2.15 Results and discussion: Line 183-186. It looks like it's part of Materials and Methods.

Answer: We have relocated the information to the “Materials and Methods” section, more specifically to the subsection 2.2 on Satellite Data Acquisition and Processing.

2.16 Results and discussion: Line 228-230. It looks like it's part of Materials and Methods.

Answer: While details are provided in the Materials and Methods section, we just briefly summarize the information to provide a contextual introduction to Figure 5 (now Figure 4).

2.17 Conclusions: Line 358-362. It seems like part of the discussion of results.

Answer: It has been moved to “Results and discussion” section as recommended.

2.18 Conclusions: I suggest restructuring the conclusions. Conclude based on the objective and in a more concrete way.

Answer: We find the Conclusion section to be concise, comprising essentially two paragraphs. The first paragraph outlines the accuracies of the best models, while the second highlights the optimal vegetation indexes and forecast time periods. Furthermore, two additional paragraphs suggest future studies to independently validate the proposed models at different levels - farm, state, and country - and to explore the potential application of the proposed procedure to other crops in the Brazilian Cerrado.

Reviewer 3 Report

Comments and Suggestions for Authors

The article entitled ''Regional-scale cotton yield estimate in the Brazilian Cerrado using simple linear regression models with time-series of vegetation indices from EOS-MODIS satellite data'' performs a simple linear regression approach between measured cotton productivity data and vegetation indices in several cotton fields spread across the Brazilian Cerrado. The article is of high quality, in line with the journal, but requires adjustments in all sections. The authors do not differentiate between the keywords in the title in the title and avoid confusing the MODIS sensor with the satellite. Please remove the review Table 1 from the introduction and transform it into an introduction text, in addition to complementing the discussion of the results. It would be imperative to mention some types of cotton cultivars used in the research. It is necessary to improve the resolution of the figure that shows the location of the study areas, including geographic coordinates and highlighting Brazil's position in South America. Figure 1 presented is small. The correlations include the significance level for 95 or 99%, for example, the p-values, which would be imperative to add. The error in estimating cotton productivity is considerable (approximately 740 kg/ha). It would be interesting to calculate and present this value as a percentage. In addition to pixel size, discuss other factors that influence errors and suggest improvements to estimates. The conclusion agrees with the goals but removes unnecessary discussions. In the document, words and expressions in blue refer to the lack of commas, articles, or modification suggestions. The words underlined in red are associated with suggestions to improve the passive voice and change unclear words and phrases. Follow the improvement suggestions mentioned above and below to improve the article.

Article notes:

24. Most recent forecasting methods... Give examples.

26. ...for big data, and their inherent... add a comma after data

27- 30. This paper proposes an approach based on simple linear models... Specify the type of approach. Add regression to simple linear model. MODIS is not a satellite. Specify the regular interval...

32. ... averaged in 15-day intervals. Change 15-days to 15-day intervals

35. remove the from... were from the 90-105 to 135...

36. add a comma in ...open boll, and...

add the ... maturation, with the lowest RMSE of about...

38. ... of the VI time-series) for EVI and.. change to time series

40. add a comma after ...contour maps, and... Change its to their.

41. Replace the words: yield prediction, cotton, Brazilian Cerrado. They are part of the text.

23 to 42. In red, improve: wordy sentences, passive voice, word choice, comma misuse within clauses.

Introduction

43-62. Improve: wordy sentences, word choice, passive voice misuse, comma misuse clauses.

68. Remove the table from the introduction and transform it into text. There is a lot of information that can be used in discussions.

63-96 – Punctuation in compound/complex sentences, word choice, passive voice misuse, potentially sensitive language, wordy sentences.

103. Neither EOS nor MODIS are satellites. The satellites are Terra and Acqua.

105. .... from several MODIS spectral bands… Specify the spectral range of the MODIS sensor bands.

106. These regular interval averages... Not clear! To improve.

102-110. Improve writing.

113 – 120. Improve: add comma, missing words, change words ...is depicted by depicts. Passive voice misuse, outdated language, and wordy sentences.

121. Figure 1 has poor visualization quality. The state acronyms are without identification. It is not possible to understand what the selected clippings are. Suggestion: presents the map of South America with the image of Brazil and the states within it. Shows areas with a zoom.

125-139. Improve writing. Punctuation, passive voice misuse and word choice.

127. In Figure 2 are shown histograms… Figure 2 represents result and not material and method. This is an exploratory data analysis. Explain how these results were produced.

In Figure 2 are shown histograms for some of these parameters. Specify the parameters. What is the purpose of showing these histograms? Exploratory data analysis! Results.

The average seed cotton yield… Specify the cotton cultivar.

141. Scroll to Figure 2 for results and discussion.

146. ..the closest climate station to each… Change by weather station. Indicate whether it is conventional or automatic?

152-179. Correct: Punctuation, passive voice misuse, word choice and unclear sentences.

156. Insert reference for the MODIS product.

162. Insert reference to Google Earth Engine.

207-226 – Improve punctuation, passive voice misuse, word choice.

208. Reflectance of NIR... Add the before reflectance. The reflectance...

217. add before negative the word a. Thenkabail et al. [33] obtained a negative...

233. Time-series... change to The time series...

228-254. Improve punctuation, passive voice misuse, word choice, nuclear sentences.

224. Indicate in the figure caption who the graph is b)....

265. Test the significance level of R2 at the 95 or 99% level?

268-288. Improve: Improve punctuation, passive voice misuse, word choice.

300-325. Improve: Improve punctuation, passive voice misuse, word choice, text inconsistencies.

303. Figure 3. Specify the locations where the images were taken.

332-353. Improve: Improve punctuation, passive voice misuse, word choice,

358-361. The conclusion cannot contain discussion. Remove to the results and discussion section.

Comments on the Quality of English Language

The article entitled ''Regional-scale cotton yield estimate in the Brazilian Cerrado using simple linear regression models with time-series of vegetation indices from EOS-MODIS satellite data'' performs a simple linear regression approach between measured cotton productivity data and vegetation indices in several cotton fields spread across the Brazilian Cerrado. The article is of high quality, in line with the journal, but requires adjustments in all sections. The authors do not differentiate between the keywords in the title in the title and avoid confusing the MODIS sensor with the satellite. Please remove the review Table 1 from the introduction and transform it into an introduction text, in addition to complementing the discussion of the results. It would be imperative to mention some types of cotton cultivars used in the research. It is necessary to improve the resolution of the figure that shows the location of the study areas, including geographic coordinates and highlighting Brazil's position in South America. Figure 1 presented is small. The correlations include the significance level for 95 or 99%, for example, the p-values, which would be imperative to add. The error in estimating cotton productivity is considerable (approximately 750 kg/ha). It would be interesting to calculate and present this value as a percentage. In addition to pixel size, discuss other factors that influence errors and suggest improvements to estimates. The conclusion agrees with the goals but removes unnecessary discussions. In the document, words and expressions in blue refer to the lack of commas, articles, or modification suggestions. The words underlined in red are associated with suggestions to improve the passive voice and change unclear words and phrases. Follow the improvement suggestions mentioned above and below to improve the article.

Article notes:

24. Most recent forecasting methods... Give examples.

26. ...for big data, and their inherent... add a comma after data

27- 30. This paper proposes an approach based on simple linear models... Specify the type of approach. Add regression to simple linear model. MODIS is not a satellite. Specify the regular interval...

32. ... averaged in 15-day intervals. Change 15-days to 15-day intervals

35. remove the from... were from the 90-105 to 135...

36. add a comma in ...open boll, and...

add the ... maturation, with the lowest RMSE of about...

38. ... of the VI time-series) for EVI and.. change to time series

40. add a comma after ...contour maps, and... Change its to their.

41. Replace the words: yield prediction, cotton, Brazilian Cerrado. They are part of the text.

23 to 42. In red, improve: wordy sentences, passive voice, word choice, comma misuse within clauses.

Introduction

43-62. Improve: wordy sentences, word choice, passive voice misuse, comma misuse clauses.

68. Remove the table from the introduction and transform it into text. There is a lot of information that can be used in discussions.

63-96 – Punctuation in compound/complex sentences, word choice, passive voice misuse, potentially sensitive language, wordy sentences.

103. Neither EOS nor MODIS are satellites. The satellites are Terra and Acqua.

105. .... from several MODIS spectral bands… Specify the spectral range of the MODIS sensor bands.

106. These regular interval averages... Not clear! To improve.

102-110. Improve writing.

113 – 120. Improve: add comma, missing words, change words ...is depicted by depicts. Passive voice misuse, outdated language, and wordy sentences.

121. Figure 1 has poor visualization quality. The state acronyms are without identification. It is not possible to understand what the selected clippings are. Suggestion: presents the map of South America with the image of Brazil and the states within it. Shows areas with a zoom.

125-139. Improve writing. Punctuation, passive voice misuse and word choice.

127. In Figure 2 are shown histograms… Figure 2 represents result and not material and method. This is an exploratory data analysis. Explain how these results were produced.

In Figure 2 are shown histograms for some of these parameters. Specify the parameters. What is the purpose of showing these histograms? Exploratory data analysis! Results.

The average seed cotton yield… Specify the cotton cultivar.

141. Scroll to Figure 2 for results and discussion.

146. ..the closest climate station to each… Change by weather station. Indicate whether it is conventional or automatic?

152-179. Correct: Punctuation, passive voice misuse, word choice and unclear sentences.

156. Insert reference for the MODIS product.

162. Insert reference to Google Earth Engine.

207-226 – Improve punctuation, passive voice misuse, word choice.

208. Reflectance of NIR... Add the before reflectance. The reflectance...

217. add before negative the word a. Thenkabail et al. [33] obtained a negative...

233. Time-series... change to The time series...

228-254. Improve punctuation, passive voice misuse, word choice, nuclear sentences.

224. Indicate in the figure caption who the graph is b)....

265. Test the significance level of R2 at the 95 or 99% level?

268-288. Improve: Improve punctuation, passive voice misuse, word choice.

300-325. Improve: Improve punctuation, passive voice misuse, word choice, text inconsistencies.

303. Figure 3. Specify the locations where the images were taken.

332-353. Improve: Improve punctuation, passive voice misuse, word choice,

358-361. The conclusion cannot contain discussion. Remove to the results and discussion section.

Author Response

Reviewer #3:

3.1 Comments and Suggestions for Authors: The article entitled ''Regional-scale cotton yield estimate in the Brazilian Cerrado using simple linear regression models with time-series of vegetation indices from EOS-MODIS satellite data'' performs a simple linear regression approach between measured cotton productivity data and vegetation indices in several cotton fields spread across the Brazilian Cerrado. The article is of high quality, in line with the journal, but requires adjustments in all sections. The authors do not differentiate between the keywords in the title in the title and avoid confusing the MODIS sensor with the satellite. Please remove the review Table 1 from the introduction and transform it into an introduction text, in addition to complementing the discussion of the results. It would be imperative to mention some types of cotton cultivars used in the research. It is necessary to improve the resolution of the figure that shows the location of the study areas, including geographic coordinates and highlighting Brazil's position in South America. Figure 1 presented is small. The correlations include the significance level for 95 or 99%, for example, the p-values, which would be imperative to add. The error in estimating cotton productivity is considerable (approximately 740 kg/ha). It would be interesting to calculate and present this value as a percentage. In addition to pixel size, discuss other factors that influence errors and suggest improvements to estimates. The conclusion agrees with the goals but removes unnecessary discussions. In the document, words and expressions in blue refer to the lack of commas, articles, or modification suggestions. The words underlined in red are associated with suggestions to improve the passive voice and change unclear words and phrases. Follow the improvement suggestions mentioned above and below to improve the article.

Answer: Answers for all issues above are addressed individually bellow.

3.2 Abstract: Line 24. Most recent forecasting methods... Give examples;

Answer: Examples included (Random Forest and Artificial Neural Networks)

3.3 Abstract: Line 26. ...for big data, and their inherent... add a comma after data

Answer: We have inserted it.

3.4 Abstract: Lines 27- 30. This paper proposes an approach based on simple linear models... Specify the type of approach. Add regression to simple linear model. MODIS is not a satellite. Specify the regular interval...

Answer: We rewrite to: “We have rewritten the sentence: “This paper presents a simpler approach, employing linear regression models fitted from time-series intervals of vegetation indices (VIs) extracted from MODIS sensor data on the Terra and Aqua satellites.” The regular interval (15-days) is specified in the preceding sentences in the Abstract.”

3.5 Abstract: Line 32. ... averaged in 15-day intervals. Change 15-days to 15-day intervals

Answer: We have made changes throughout the manuscript.

3.6 Abstract: Line 35. remove the from... were from the 90-105 to 135...

Answer: We have made the necessary corrections.

3.7 Abstract: Line 36. add a comma in ...open boll, and... add the ... maturation, with the lowest RMSE of about...

Answer: We have made the necessary corrections.

3.8 Abstract: Line 38. ... of the VI time-series) for EVI and.. change to time series

Answer: We have made changes throughout the manuscript.

3.9 Abstract: Line 40. add a comma after ...contour maps, and... Change its to their.

Answer: We have made the necessary corrections.

3.10 Keywords: Line 41. Replace the words: yield prediction, cotton, Brazilian Cerrado. They are part of the text.

Answer: Keywords was updated to: remote sensing, cotton production forecast, Brazil

3.11 Abstract: Lines 23 to 42. In red, improve: wordy sentences, passive voice, word choice, comma misuse within clauses.

Answer: We have made the necessary corrections.

3.12 Introduction: Lines 43-62. Improve: wordy sentences, word choice, passive voice misuse, comma misuse clauses.

Answer: We have made the necessary corrections.

3.13 Introduction: Line 68. Remove the table from the introduction and transform it into text. There is a lot of information that can be used in discussions.

Answer: Indeed, our research team shares the same vision as the reviewer. On the other hand, the exception for that is justified by the power to elucidate and easily compare multiple numerical and categorical data to justify and support the research. Additionally, we checked recent publications from the journal and found that this practice has been common and acceptable (e.g., Ennatiqi et al., 2024, 6, 724-753; Chaomuang et al., 2023, 5, 1865-1878; Al Kindi et al., 2023, 5, 2349-2365).

3.14 Introduction: Lines 63-96 – Punctuation in compound/complex sentences, word choice, passive voice misuse, potentially sensitive language, wordy sentences.

Answer: We have made the necessary corrections.

3.15 Introduction: Line 103. Neither EOS nor MODIS are satellites. The satellites are Terra and Acqua.

Answer: The sentence was revised: “The current study assesses the feasibility of employing simple linear models to correlate cotton yield with VIs derived from satellite data obtained from the MODIS (Moderate Resolution Imaging Spectroradiometer) sensors aboard the Terra and Aqua satellites.”

3.16 Introduction: Line 105. .... from several MODIS spectral bands… Specify the spectral range of the MODIS sensor bands.

Answer: This information was presented in the Material and Methods section.

3.17 Introduction: Line 106. These regular interval averages... Not clear! To improve.

Answer: It was rewritten to: “Reflectance time series data from multiple MODIS spectral bands were extracted to assess 15-day interval averages spanning from sowing to harvest, of widely recognized VIs.”

3.18 Introduction: Lines 102-110. Improve writing.

Answer: We have made the necessary corrections.

3.19 Material and Methods: Lines 113-120. Improve: add comma, missing words, change words ...is depicted by depicts. Passive voice misuse, outdated language, and wordy sentences.

Answer: We have made the necessary corrections.

3.20 Material and Methods: Line 121. Figure 1 has poor visualization quality. The state acronyms are without identification. It is not possible to understand what the selected clippings are. Suggestion: presents the map of South America with the image of Brazil and the states within it. Shows areas with a zoom.

Answer: States' abbreviations (MT: Mato Grosso, GO: Goiás, BA: Bahia) are now indicated in the figure legend. We attempted to include a Brazil map within the South America map in a multilevel perspective of magnification, but the resulting map became too small, further hindering visualization. For reference, plots 1, 2, and 3 (examples at each state) are now indicated on the map.

3.21 Material and Methods: Lines 125-139. Improve writing. Punctuation, passive voice misuse and word choice.

Answer: We have made the necessary corrections.

3.22 Material and Methods: Line 127. In Figure 2 are shown histograms… Figure 2 represents result and not material and method. This is an exploratory data analysis. Explain how these results were produced.

Answer: Histograms were included in Figure 2 to illustrate the distribution of these parameters. However, as the average values are described in the text, Figure 2 has been removed.

3.23 Material and Methods: In Figure 2 are shown histograms for some of these parameters. Specify the parameters. What is the purpose of showing these histograms? Exploratory data analysis! Results.

Answer: See 3.22.

3.24 Material and Methods: The average seed cotton yield… Specify the cotton cultivar.

Answer: Information about cotton cultivars used in this study was included in the Material and Methods section.

3.25 Material and Methods: Line 141. Scroll to Figure 2 for results and discussion.

Answer: See 3.22.

3.26 Material and Methods: Line 146. ..the closest climate station to each… Change by weather station. Indicate whether it is conventional or automatic?

Answer: Corrected as suggested (weather station). Both conventional and automatic weather station are employed to collect data on the AGRITEMPO platform.

3.27 Material and Methods: Lines 152-179. Correct: Punctuation, passive voice misuse, word choice and unclear sentences.

Answer: We have made the necessary corrections.

3.28 Material and Methods: Line 156. Insert reference for the MODIS product.

Answer: Reference [31] inserted.

3.29 Material and Methods: Line 162. Insert reference to Google Earth Engine.

Answer: Reference [32] inserted.

3.30 Results and Discussion: Lines 207-226 – Improve punctuation, passive voice misuse, word choice.

Answer: We have made the necessary corrections.

3.31 Results and Discussion: Line 208. Reflectance of NIR... Add the before reflectance. The reflectance...

Answer: Added.

3.32 Results and Discussion: Line 217. add before negative the word a. Thenkabail et al. [33] obtained a negative...

Answer: Added.

3.33 Results and Discussion: Line 233. Time-series... change to The time series...

Answer: Changed.

3.34 Results and Discussion: Lines 228-254. Improve punctuation, passive voice misuse, word choice, nuclear sentences.

Answer: We have made the necessary corrections.

3.35 Results and Discussion: Line 224. Indicate in the figure caption who the graph is b)....

Answer: We have made the necessary corrections.

3.36 Results and Discussion: Line 265. Test the significance level of R2 at the 95 or 99% level?

Answer: Answer:  Indeed, the analysis of the linear models requires ensuring the assumptions and getting significance for model parameters (analysis of variance). We appreciate the note on an important aspect. We included the ANalysis of Variance (ANOVA) in Table 3 to highlight the significance of the adjusted model with a p-value column. It is also essential to highlight that our approach was evaluated with a robust cross-validation method (train: 60% /  test: 40%), with relevant results. In general, when a model does not represent the modeled phenomenon, validation results show high errors disqualifying it for the task, which was not the case for the best results we showed in the article.

3.37 Results and Discussion: Lines 268-288. Improve: Improve punctuation, passive voice misuse, word choice.

Answer: We have made the necessary corrections.

3.38 Results and Discussion: Lines 300-325. Improve: Improve punctuation, passive voice misuse, word choice, text inconsistencies.

Answer: We have made the necessary corrections.

3.39 Results and Discussion: Line 303. Figure 3. Specify the locations where the images were taken.

Answer: The data primarily originated from commercial cotton farms, and the farmers did not grant permission to disclose the exact geographic coordinates of the locations.

3.40 Results and Discussion: Lines 332-353. Improve: Improve punctuation, passive voice misuse, word choice,

Answer: We have made the necessary corrections.

3.41 Conclusions: Lines 358-361. The conclusion cannot contain discussion. Remove to the results and discussion section.

Answer: We have made the necessary corrections.